# Comparison of Pan-Lyssavirus RT-PCRs and Development of an Improved Protocol for Surveillance of Non-RABV Lyssaviruses

**DOI:** 10.3390/v15030680

**Published:** 2023-03-04

**Authors:** Petra Drzewnioková, Sabrina Marciano, Stefania Leopardi, Valentina Panzarin, Paola De Benedictis

**Affiliations:** 1FAO Reference Center for Rabies, Istituto Zooprofilattico Sperimentale delle Venezie, 35020 Legnaro, PD, Italy; 2Innovative Virology Laboratory, Research and Innovation Department, Istituto Zooprofilattico Sperimentale delle Venezie, 35020 Legnaro, PD, Italy

**Keywords:** pan-lyssavirus, divergent lyssaviruses, rabies molecular diagnosis

## Abstract

Rabies is a zoonotic and fatal encephalitis caused by members of the *Lyssavirus* genus. Among them, the most relevant species is *Lyssavirus rabies*, which is estimated to cause 60,000 human and most mammal rabies deaths annually worldwide. Nevertheless, all lyssaviruses can invariably cause rabies, and therefore their impact on animal and public health should not be neglected. For accurate and reliable surveillance, diagnosis should rely on broad-spectrum tests able to detect all known lyssaviruses, including the most divergent ones. In the present study, we evaluated four different pan-lyssavirus protocols widely used at an international level, including two real-time RT-PCR assays (namely LN34 and JW12/N165-146), a hemi-nested RT-PCR and a one-step RT-PCR. Additionally, an improved version of the LN34 assay ((n) LN34) was developed to increase primer–template complementarity with respect to all lyssavirus species. All protocols were evaluated in silico, and their performance was compared in vitro employing 18 lyssavirus RNAs (encompassing 15 species). The (n) LN34 assay showed enhanced sensitivity in detecting most lyssavirus species, with limits of detection ranging from 10 to 100 RNA copies/µL depending on the strain, while retaining high sensitivity against *Lyssavirus rabies*. The development of this protocol represents a step forward towards improved surveillance of the entire Lyssavirus genus.

## 1. Introduction

Lyssaviruses (family *Rhabdoviridae*, subfamily *Alpharhabdovirinae*, genus *Lyssavirus*) are neurotropic pathogens transmissible to mammals and able to cause rabies, a zoonotic encephalitis that is almost always fatal following the onset of its symptoms [1]. *Lyssavirus rabies* includes all rabies viruses (RABV) responsible for most human and animal cases worldwide. At a global level, rabies is responsible for an estimated 60,000 human deaths per year, mostly caused by RABV transmitted by dogs in Africa and Asia [2]. RABV is a multi-host pathogen that has established independent transmission cycles in several mammalian species, thus occupying different geographical and ecological niches [3,4]. Robust disease surveillance relying on an accurate diagnostic framework contributes not only to defining the real burden of the infection in endemic areas, but also to monitor progress on control plans, as well as to detect imported cases in rabies-free areas in a timely manner. With the objective of stopping dog-mediated human rabies deaths by 2030, disease surveillance and improved diagnostics have been recognized as paramount activities within the ZeroBy30 Global Strategic Plan [2].

In addition to *Lyssavirus rabies*, sixteen other lyssavirus species are currently recognized as paramount by the International Committee on the Taxonomy of Viruses (ICTV) with two additional isolates awaiting for official classification, namely, Kotalahti bat lyssavirus (KBLV) and Matlo bat lyssavirus (MBLV) [5,6]. Members of the *Lyssavirus* genus are operationally divided into phylogroups based on genetic and antigenic differences [7]. The most divergent species, *Lyssavirus ikoma*, *Lyssavirus lleida*, *Lyssavirus caucasicus* and probably the putative MBLV, form phylogroup III [7,8]. Phylogroup I includes *Lyssavirus rabies*, *Lyssavirus aravan*, *Lyssavirus australis*, *Lyssavirus bokeloh*, *Lyssavirus duvenhage*, *Lyssavirus formosa*, *Lyssavirus hamburg*, *Lyssavirus helsinki*, *Lyssavirus gannoruwa*, *Lyssavirus irkut*, *Lyssavirus khujand* and probably the putative KBLV [9]. Phylogroup II includes the African *Lyssavirus lagos*, *Lyssavirus mokola* and *Lyssavirus shimoni*.

Unlike RABV, each non-RABV lyssavirus has most likely co-evolved with a specific host, mainly one or two sibling bats per viral species, and presents a restricted geographical range linked to the ecology of the hosts. Despite rare spillover events in non-flying mammals, no forward transmission or subsequent establishment in terrestrial mammals has been reported so far [3]. Although the risk of spillover events is low due to the history of co-evolution of lyssaviruses with their hosts, their impact on animal and human health is considerable and should not be neglected [10,11,12,13,14]. For this reason, control and prevention measures, whenever available, must be equally guaranteed in case of human and domestic animal exposure to non-RABV lyssaviruses [8].

Reliable and rapid diagnostic tests are a prerequisite both for monitoring rabies distribution in animal reservoirs and for identifying spillover cases in both humans and animals. Rabies diagnosis in animals is performed post-mortem through the identification of the etiological agent (nucleic acid or abundant proteins) from the brain tissue. Until recently, such a procedure was largely based on methods targeting the viral antigens (i.e., the gold-standard fluorescent antibody (DFA) test or the more recent direct rapid immunochemical test, DRIT) [15,16,17]. Reverse transcription polymerase chain reaction (RT-PCR) has only recently been included as a standard technique for rabies diagnosis in animals [16], although the DFA test had been complemented by several molecular methods in recent years [17]. Nowadays, international organizations agree in recommending the use of a pan-lyssavirus molecular method for diagnostic purposes [15,16,18]; such assays should be designed to allow the detection of the broadest spectrum of lyssaviruses, including those expected to be rarely causing infections in humans and domestic animals [10,12]. Molecular protocols should meet the international criteria for validation, including a comparison of performances against the gold-standard antigen-based methods [16,19]. It is important to highlight that gold-standard methods for rabies detection may have lower sensitivity compared to molecular techniques with respect to the detection of non-RABV lyssaviruses. This is not only due to an obvious lower sensitivity of antigen-based vs. molecular-based methods [20], but also to the fact that enlarging the inclusivity of the available DFA/DRIT conjugates, which were originally developed for the detection of RABV, implies a huge effort currently not addressed by manufacturers [21,22]. Nevertheless, suboptimal detections and diagnostic dropouts might in turn reduce the effectiveness of surveillance, hampering the implementation of prompt mitigation measures.

To promote better the surveillance of RABV and non-RABV viruses, we aimed at identifying a broadly reactive molecular method able to detect different lyssavirus phylogroups with high sensitivity. To this end, we evaluated, in silico and in vitro, three WHO/WOAH recommended assays (namely, the LN34 and JW12/N165-146 real-time RT-PCRs (rRT-PCRs) and the hemi-nested RT-PCR developed by Heaton et al.) and a well-established one-step RT-PCR protocol employed at our laboratory and deployed at an international level [18,23,24,25,26]. The JW12/N165-146 assay was originally developed as a TaqMan lyssavirus assay using specific probes to differentiate among RABV, EBLV-1 and EBLV-2 lyssavirus species [23,24]. The RABV-specific protocol was evaluated using a large panel of field samples representative of the RABV diversity across the globe, displaying failure in detecting several RABV strains [27,28]. To overcome the low inclusivity of the probes, the original protocol was further adapted to the intercalating dye SYBR^®^ Green [29], and showed an ability to detect all lyssaviruses known at that time with better sensitivity than the hemi-nested RT-PCR. The protocol evaluated in our study was the one further modified as one-step rRT-PCR (JW12/N165-146) [30]. The LN34 assay recommended by the WHO [18,26,31] uses a combination of two forward and one reverse primer and two locked nucleic acid (LNA) probes designed in the same highly conserved region targeted by the JW12 primer. The LN34 probes demonstrated high tolerance to single-nucleotide polymorphisms (SNPs), showing optimal performances [31] and more inclusivity than JW12/N165-146, as noticed elsewhere [28]. Probes and primers were designed based on 13 lyssavirus species, excluding LLEBV and GBLV, in addition to others yet to be discovered [31]. In this context, it is worth mentioning that, while LN34’s diagnostic sensitivity was extensively addressed using different sample types and viral species, its analytical sensitivity was assessed on an artificial RNA (127 bp long) and total RNA from different RABV-, MOKV- and DUVV-cell-adapted strains [26,31].

Based on the promising high inclusivity of the LN34 probes, we further investigated the potential of an improved version of the LN34 assay (later referred as (n) LN34) by developing a new and broadly reactive set of primers able to recognize all known lyssavirus species with high sensitivity.

## 2. Materials and Methods

### 2.1. In Silico Evaluation

We tested the inclusivity of reference protocols targeting the 3′UTR and/or N gene of *Lyssavirus* (Table 1) in silico using Geneious Prime^®^ 2022.1.1 (Biomatters, Auckland, New Zealand). To this end, a multiple nucleotide sequence alignment (MSA) covering the assay target regions was created using 269 lyssavirus sequences available in GenBank that encompass all the lyssaviruses described so far. The MSA is available in Appendix A.

For all the sequences in the MSA and for each assay under evaluation, we assessed the total number of mismatches (TNMM) by summing the number of mismatches occurring in sense and antisense primers. In the case of assays with two or more oligonucleotides annealing the same target region (reverse primers for conventional RT-PCR and forward primers and probes for LN34), we took into consideration the number of mismatches related to the best-fitting oligonucleotide. The mean of the TNMM (ranging from 0 to 7) was then mathematically calculated for each assay against different lyssavirus species. In addition, the percentage of sequences showing the same TNMM was calculated within each phylogroup. We set a threshold of TNMM ≤ 2 as an acceptable value for test functionality with sufficient sensitivity.

### 2.2. Improvement of LN34 rRT-PCR Assay

Based on in silico testing results, we modified the LN34 assay primers to increase coverage across lyssaviruses. In detail, degenerated nucleotides were included to accommodate all phylogroups with a value of TNMM ≤ 2, in order to guarantee sufficient sensitivity and inclusivity. To avoid a high number of degenerated nucleotides per primer that could impair assay functionality, we pooled different primers with a limited number of degenerated sites to match the observed polymorphisms. Primers were designed using “primer3” (Geneious Prime^®^ 2022.1.1, Biomatters, Auckland, New Zealand) to satisfy requirements regarding the melting temperature (i.e., for original primers), absence of secondary structures and mismatches in proximity of the 3′ end.

### 2.3. Sample Panel Selection

In total, 18 RNA transcripts (Table 2) were produced to compare the performances of the protocols tested in this study, including most lyssavirus species from phylogroup I and all divergent species from phylogroups II and III. For RABV, we selected two strains from the Cosmopolitan and the Africa 2 lineages.

Plasmids containing lyssavirus genome sequences encompassing assay target regions were produced in-house using the pCR™II Vector (TA Cloning™ Kit, Dual Promoter, Invitrogen, MA, USA) or purchased from Vector Builder (Neu-Isenburg, Germany), based on the reference sequences available in GenBank. In vitro-transcribed RNA was produced from plasmid templates, and its quality, purity and concentration were determined as described in Appendix A.

In vitro-transcribed RNA was aliquoted, supplemented with 40 U of RNasin Plus RNase Inhibitor (Promega) and stored at ≤−70 °C until use.

### 2.4. Comparison of Lyssavirus Detection Methods

To assess the performance of different molecular protocols, we determined the limit of detection (LoD) using ten-fold dilution series of RNA transcripts produced as above. To simulate clinical samples, dilutions of transcripts were prepared using a batch of RNA isolated from pooled mammalian brains (cats, dogs, foxes, martens, badgers, wolves and various species of bats) that had previously tested negative for lyssaviruses during routine surveillance activities. All the dilutions were tested in triplicate on the same day. The JW12/N165-146 (performed as one-step rRT-PCR based on the intercalating dye SYBR^®^ Green) and the probe-based LN34 assay were carried out as described in *Laboratory techniques in rabies*, volume 2 (WHO) [18]. An exception was made for the final concentration of primers of the LN34 assay, which was increased from 10 to 20 µM based on the amendments suggested by the authors (courtesy of C. Gigante). The novel protocol developed in this study, the (n) LN34 assay (detailed protocol in Appendix A), differs from the original LN34 assay in the primer mix only. Upon observation of the poor performance of the hemi-nested RT-PCR [18], the protocol was modified employing the Platinum Taq DNA Polymerase (Invitrogen) to improve sensitivity. The one-step RT-PCR was carried out as described previously [18,25]. All rRT-PCRs [18,26] were run on a Bio-Rad CFX96 Touch System and analyzed using the Bio-Rad CFX Manager software (Version 3.1). All RT-PCRs were carried out in Bio-Rad Thermal cycler, while GeneAmp PCR System 9700 (Applied Biosystems) was used when a higher reaction volume (50 µL) needed to be accommodated. Amplification products were visualized with the QIAxcel Advanced System (Qiagen).

The limit of detection (LoD) was determined as the highest dilution at which all tested replicates yielded positive results. For LN34 and (n) LN34, Ct values up to 35.9 were accepted. LoD determination for JW12/N165-146 was also based on the presence of peaks with the expected melting temperature. For hemi-nested and one-step RT-PCR, the LoD was determined as the highest dilution at which all tested replicates yielded positive results as evaluated via visualization of the amplicons after gel electrophoresis; the viral identity was confirmed using Sanger sequencing.

### 2.5. Repeatability and Analytical Specificity (ASp) of (n) LN34

Repeatability was assessed by testing low (LoD) and medium (the LoD + 2 log) RNA transcript concentrations for each phylogroup employing EBLV-1 a (phylogroup I), LBV a (phylogroup II) and IKOV (phylogroup III). Samples were tested in triplicate on two different days by one operator using freshly prepared RNA dilutions.

To determine (n) LN34 exclusivity and absence of a cross-reaction against microorganisms other than lyssaviruses, 18 nucleic acids from non-target viruses or bacteria (*West Nile virus* lineage 1, *West Nile virus* lineage 2, *Usutu virus*, *Equid alphaherpesvirus* 1, *Equid alphaherpesvirus* 4, *Human astrovirus* 1, *Canine astrovirus*, *Mamastrovirus* 3, *Tick-borne encephalitis virus* strain Hypr, *Canine distemper virus*, *Avian influenza virus* H5N1 HPAI, *Lloviu virus*, SARS-CoV-2, *Leptospira icterohaemorragiae*, *Listeria innocua* and *monocytogenes*, *Escherichia coli* and *Staphylococcus* spp.) were tested. Selectivity against host matrix components was also evaluated employing RNA extracted from the non-infected brain of 16 mammal species *(Canis aureus*, *Canis lupus familiaris*, *Canis lupus*, *Capreolus capreolus*, *Equus asinus*, *Felis catus*, *Hypsugo savii*, *Meles meles*, *Miniopterus schreibersii*, *Mus musculus*, *Pipistrellus kuhlii*, *Pipistrellus nathusii*, *Pipistrellus pipistrellus*, *Plecotus auritus*, *Plecotus auritus*, *Vulpes vulpes)*. All samples were tested in duplicate.

## 3. Results

### 3.1. In Silico Evaluation

The MSA included 269 nucleotide sequences of 17 established lyssaviruses species and 2 putative species. More specifically, 251 sequences belonged to phylogroup I (of which, 122 were RABV), 12 to phylogroup II, and 6 to phylogroup III.

Appendix A itemizes the annealing regions of the oligonucleotide sets under evaluation, with respect to representative sequences of all known lyssavirus species. The primer pair from the first round of the hemi-nested RT-PCR showed a TNMM of ≤2 with 100, 91 and 83% of sequences from phylogroup I, phylogroup II and phylogroup III, respectively, while a TNMM >2 was seen in 9% and 17% of sequences from phylogroups II and III, respectively. A primer pair from the second round showed a TNMM of ≤2 in all sequences from phylogroups I and II, and a TNMM of >2 in 27% of sequences from phylogroup III. The one-step RT-PCR set of primers showed acceptable complementarity with 100% of sequences belonging to phylogroups I and III and with 83% of sequences belonging to phylogroup II. The JW12/N165-146 set of primers exhibited a TNMM of ≤2 with 97, 50 and 83% of sequences belonging to phylogroups I, II and III, respectively. As for the LN34 assay, the probes achieved acceptable complementarity with all 269 sequences belonging to the Lyssavirus genus. In contrast, a TNMM of >2 against LN34 primers occurred in 17 and 25% of sequences from phylogroups I and II, respectively. Notably, a TNMM of 6 to 7 occurred in 50% of the available sequences of phylogroup III (i.e., 3/6). Further details are presented in Table 3, and the mean TNMM between all primer and probe sets for each lyssavirus is shown in Figure 1.

The complete list of sequences and the minimal number of mismatches observed between each set of primers/probes and target sequence are presented in Appendix A.

### 3.2. (n) LN34 Assay Development

Probe-based rRT-PCR assays are known to have higher sensitivity, reproducibility and specificity, with no post-PCR processing and a shorter turnaround time. For these reasons, and taking into consideration LN34 probes’ high complementarity and tolerance to single-nucleotide polymorphisms (SNPs) with respect to the entire lyssavirus genus, we developed an improved version of this assay, producing a new set of primers capable of matching all *Lyssavirus* species known at the time of writing. In detail, primers FW1/FW2/FW3, FW4/REV2 and FW5/REV3 were designed to enhance the detection of phylogroup I, LLEBV/IKOV and WCBV/MBLV, respectively. The FW5/REV3 re-design was based on a modification of a previously published species-specific rRT-PCR [10]. As a requirement, primers were designed to exhibit a TNMM of ≤ 2. REV1 and probes were as per the original paper [26] (Table 4).

As shown in Figure 1, when compared to the original protocol, the new primer combination (Table 4) resulted in a higher coverage with respect to all lyssaviruses, with the exception of SHIBV displaying three mismatches.

In detail, the new set of primers increased inclusivity against all representative sequences from both phylogroups I (83% vs. 100%) and III (0% vs. 100%) (Table 5). For phylogroup II, although the percentage of sequences meeting the criteria for optimal complementarity remained unchanged compared to the original LN34 assay, the novel primer set allowed a reduction to one total mismatch in 17% of sequences (Table 5).

### 3.3. Analytical Sensitivity (ASe)

A comparative overview of the analytical sensitivity of all the assays under evaluation is provided in Figure 2. The hemi-nested RT-PCR showed an LoD of 10^3^–10^4^ RNA copies/µL for the majority of the RNAs tested, except MBLV and LBV C, which reached an LoD of 100 and 10^5^ RNA copies/µL, respectively. Compared to the hemi-nested RT-PCR, the one-step RT-PCR showed the same ASe in detecting 5/18 species, and higher sensitivity in detecting 11/18 lyssaviruses. With respect to real-time assays, the one-step RT-PCR performed better only for EBLV-2 (LoD = 100 RNA copies/µL vs. 1000 RNA copies/µL achieved by the LN34 and the JW12/N165-146 assays) and for SHIBV, for which the best performance among all the tested protocols (LoD = 10 RNA copies/µL) was obtained. LN34 and JW12/N165-146 rRT-PCR assays demonstrated comparable sensitivity in detecting 8/18 reference strains. Compared to other assays, LN34 showed the highest sensitivity in detecting 6/18 representative sequences, with the greatest sensitivity for LBV lineage A. The LN34 assay showed a worse performance for ARAV, LLEBV and MBLV. The JW12/N165-146 rRT-PCR had enhanced sensitivity for 4/18 strains and the greatest sensitivity for BBLV. Poorer detection using JW12/N165-146 rRT-PCR was observed for ABLV, WCBV and LBV strains (Figure 2).

The (n) LN34 assay showed enhanced sensitivity in detecting seven and eight reference strains compared to the original LN34 and the JW12/N165-146 rRT-PCR, respectively. LoDs ranged between 10 and 100 RNA copies/µL for all but SHIV and LBV lineage C, for which the LoD recorded was 1000 RNA copies/µL. Remarkably, the improved protocol displayed a dramatic increase in sensitivity compared to the original LN34 assay in detecting all phylogroups III representatives, as well as ARAV, DUVV and EBLV-2 (with differences spanning from 1 up to 5 logarithms) (Figure 2). A reduced sensitivity of 1 log with respect to the original LN34 was observed only for LBV lineage A and SHIBV (100 to 1000 RNA copies/µL). Detailed information is presented in Appendix A.

### 3.4. (n) LN34 Specificity and Repeatability

The newly designed set of primers turned out to be highly specific, yielding negative results for brain tissues of non-infected mammals (16 host species tested) and non-target viruses and bacteria (n = 18).

The within-run and between-days repeatability was ≥96.72 (Table 6), which demonstrates the high repeatability of the (n) LN34 regardless of the phylogroup tested.

## 4. Discussion

Over the years, several molecular assays have proven reliable in detecting a great variety of lyssavirus RNAs, and the use of a molecular method and an analytical framework in rabies diagnostics is currently recommended internationally both to increase sensitivity and shorten laboratory turnaround times [16,17,18]. In this context, ensuring pan-lyssavirus sensitivity against divergent lyssaviruses represents a paramount requirement in areas where such a broad diversity of lyssaviruses is expected, such as in the Eurasian and African continents.

We selected four different pan-lyssavirus RT-PCRs described in the literature, all targeting the leader sequence and/or the N gene. These highly conserved regions ensure maximum protocol sensitivity since they are transcribed in the highest abundance compared to downstream genes [32]. All these protocols are implemented at an international level by national and regional diagnostic laboratories [23,33,34,35,36], including two largely adopted one-step real-time RT-PCRs (rRT-PCR), namely JW12/N165-146 [18,24] and LN34 [26] assays, and two conventional RT-PCRs, namely hemi-nested RT-PCR [23] and one-step RT-PCR [25].

In our study, we compared the sensitivity of these protocols via combining the well-established assessment of analytical sensitivity (ASe) according to WOAH guidelines and recommendations [19] along with in silico determination of oligonucleotide complementarity against lyssavirus reference sequences available in GenBank. This approach allowed us to evaluate back-to-back the selected molecular methods using a much broader set of viruses within the *Lyssavirus* genus, including several clades/variants of RABV and lyssavirus lineages unavailable in our repository stock. Our results confirm that primer/sequence complementarity is a crucial factor in achieving acceptable sensitivity of the method. In this context, we observed that a total number of mismatches (TNMM) of three or more impaired viral detection in the rRT-PCR protocols under evaluation, which is in agreement with what has been observed by other authors [28]. Thus, we used this criterion as a guide to modify primers of the LN34 assay to widen its inclusivity and sensitivity, with particular consideration of phylogroups I and III.

We observed that the same criterion of acceptability for mismatches could not be adopted for the conventional RT-PCR protocols (hemi-nested vs. one-step), nor could we compare it with rRT-PCRs. As a matter of fact, the set of primers used for the conventional RT-PCRs demonstrated optimal complementarity with the template but performed more poorly than the rRT-PCRs. While we would in principle discourage the use of a nested protocol in diagnostic routine, due to the high risk of false-positive results and time consumption, the suboptimal performances of the hemi-nested RT-PCR observed in this study do not support its promotion for diagnostic purposes.

Overall, our study provided a comprehensive assessment of the ASe for the protocols under evaluation by using the RNA transcripts of 18 strains (representing 15 lyssavirus species), including some new ones. This approach complemented the information available in the literature, considering that the data reported for the ASe for all the protocols included in our study were limited to RABV or to only a few other *Lyssavirus* species [30,31,37,38]. Remarkably, we diluted the synthetic RNA into a matrix of RNA extracted from brain specimens to mimic, as much as possible, the broad variety of RNA inhibitors that might be present in animal specimens collected from the field.

Of note, we employed RNA transcripts as standard for multiple reasons, among them the lower biosafety level required and the reliability of the target sequence compared to cell-adapted strains. Indeed, lyssaviruses amplification not only requires biosafety level 3 laboratory facilities, but more importantly induces the fixation of genetic mutations occurring during virus adaptation to a cell substrate, which might critically affect the evaluation of the assay performance [39,40]. Consistently with this, in the framework of previous research activities, in the cell-adapted LLEBV, we observed a G11A mutation within the 3′ non-coding region where LN34/(n)LN34 forward primers bind (Appendix A). The in silico evaluation of this mutation determined a decrease in the Tm of primer FW4 that in turn might negatively affect assay sensitivity. In this context, researchers should always verify the absence of any mutations of the cell-adapted viruses before using them as standards in further validation steps. Similar mutations, rare for lyssavirus in their reservoir hosts, might be encountered, for instance, upon viral spillover and adaptation to a new host, an occurrence that should be carefully taken into consideration when interpreting the diagnostic results.

Although the original LN34 assay is applied worldwide and has undoubted benefits in terms of sensitivity with respect to RABV diagnosis, its exclusivity and ASe in detecting divergent viruses belonging to phylogroup III were the main drivers that led us to develop the (n) LN34 assay. Each of the newly designed primers incorporated a relatively small number of base degeneracies, which limited the resulting number of unique sequence combinations; the number of sequence combinations would have been much higher if single highly degenerate forward and reverse primers had been used to encompass all possible base variations (Appendix A). This design approach limited the loss of sensitivity associated with highly divergent phylogroups [8]. Consistent with what we observed in silico, primer modification of the (n) LN34 assay resulted in optimal complementarity with phylogroups I and III sequences, leading to a sharp improvement in sensitivity of up to five logarithms for divergent viruses.

Non-RABVs are generally confined to their own hosts, although they might occasionally spillover to non-host species including domestic animals and humans [10,41,42]. Although rare, these events are expected to increase due to widespread human encroachment into wildlife habitat, as testified by the identification of a group of bent-winged bats (*Miniopterus schreibersii*), a strict hypogeal bat species, and the natural host for WCBV, unexpectedly found in the city of Arezzo, Italy [10]. Cats play a crucial role as accidental hosts and effective rabies transmitters of bat-associated lyssaviruses, and this because of their free roaming and predatory behavior towards small animals, such as bats. In highly anthropized settings, bat-to-cat lyssavirus infections have been reported, such as the ones linked to EBLV-1 in France, WCBV in Italy, and also LBV in sub-Saharan Africa [10,41,43]. In this context, it is noteworthy that full typing of rabies positive cases is rarely conducted in RABV-endemic areas, which precludes quantifying the actual prevalence of non-RABV rabies encephalitis occurring in domestic animals. Thus, based on the potential consequences arising from non-RABV lyssavirus exposure in humans [10], enhanced laboratory-based surveillance with highly accurate and sensitive diagnostic methods is urgently required. This is because biting events as well as cases of syndromic animals infecting humans are expected to increase due to climate change and human encroachment.

## 5. Conclusions

Simultaneous comparisons of widely used molecular protocols for the detection of lyssavirus nucleic acid are timely and necessary. In this paper, we offered a comparative overview of the analytical sensitivity of four different methodologies and developed a valid alternative to enhance lyssavirus laboratory-based surveillance. Nonetheless, it is recommended that each diagnostic laboratory evaluate which molecular method is most appropriate for its region and needs, according to equipment availability, the experience of the staff, and the reason for testing.

In the present study, the (n) LN34 assay proved to have optimal sensitivity and inclusivity for all the known lyssaviruses; however, a continuous re-evaluation of the molecular protocols in use is needed to ensure optimal functionality over time as novel divergent strains are discovered. Finally, we suggest that assay assessments and re-evaluations should be conducted through the engagement of technical experts and the timely release of curated/revised genetic sequences of lyssaviruses identified at a global level.

## Figures and Tables

**Figure 1 viruses-15-00680-f001:**
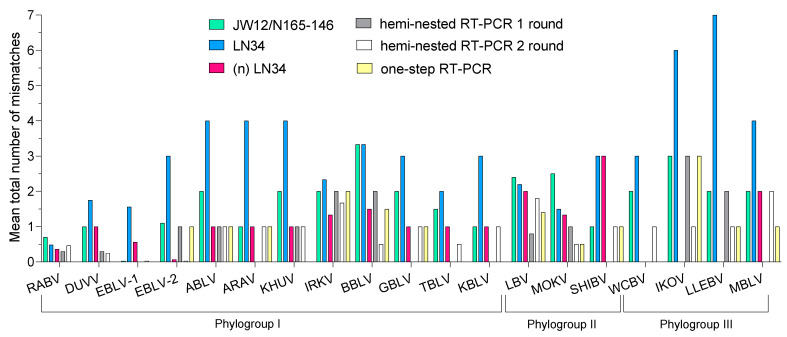
Mean total number of mismatches observed between primer sets of the assays under evaluation against 269 *Lyssavirus* sequences. This figure was created using GraphPad Prism version 9.0.

**Figure 2 viruses-15-00680-f002:**
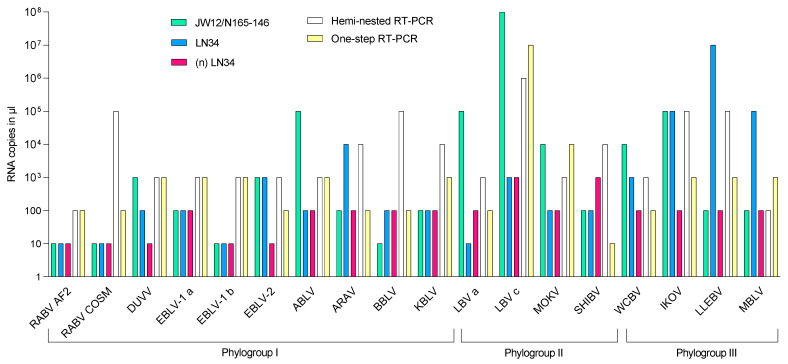
Limit of detection (LoD) of the assays under evaluation against 18 representative lyssaviruses.

**Table 1 viruses-15-00680-t001:** Reference pan-lyssavirus assays tested in silico and in vitro.

Assay	Target	Name	Type	S/As	Sequence (5′ to 3′)
		JW12	primer	S	ATGTAACACCYCTACAATG
Hemi-nested [18,23]	N gene	JW6 UNI	primer	As	ARTTVGCRCACATYTTRTG
		JW10 UNI	primer	As	GTCATYARWGTRTGRTGYTC
One-step [25]	N gene	RabForPyro	primer	S	AACACYYCTACAATGGA
		RabRev1Pyro	primer	As	TCCAATTNGCACACATTTTGTG
		RabRev2Pyro	primer	As	TCCARTTAGCGCACATYTTATG
		RabRev3Pyro	primer	As	TCCAGTTGGCRCACATCTTRTG
JW12/N165-146 [18,24]	N gene	JW12	primer	S	ATGTAACACCYCTACAATG
		N165-145	primer	As	GCAGGGTAYTTRTACTCATA
LN34 [18,26]	3′UTR and N gene	LN34 FW1	primer	S	ACGCTTAACAACCAGATCAAAGAA
		LN34 FW2	primer	S	ACGCTTAACAACAAAATCADAGAAG
		LN34Probe	probe	S	FAM-AACACCYCTACAATGGA-BHQ1
		LN34lago Probe	probe	S	FAM-AACACTACTACAATGGA-BHQ1
		LN34REV	primer	As	CMGGGTAYTTRTAYTCATAYTGRTC

S: sense, As: antisense.

**Table 2 viruses-15-00680-t002:** In vitro-transcribed RNA used in this study to compare pan-lyssavirus protocols.

Phylogroup	Species	Virus Abbreviation	Virus Name/Clade or Lineage	Genbank Accession Number
I	*Lyssavirus rabies*	RABV AF2	rabies virus*/*lineage Africa 2	MK471246
		RABV COSM	rabies virus*/*lineage Cosmopolitan	KR906742
	*Lyssavirus duvenhage*	DUVV	Duvenhage virus	EU293120
	*Lyssavirus hamburg*	EBLV-1 a	European bat lyssavirus type 1/lineage a	MF187809
		EBLV-1 b	European bat lyssavirus type 1/lineage b	MF187859
	*Lyssavirus helsinki*	EBLV-2	European bat lyssavirus type 2	KY688150
	*Lyssavirus australis*	ABLV	Australian bat lyssavirus	AY573937
	*Lyssavirus aravan*	ARAV	Aravan virus	NC_020808
	*Lyssavirus bokeloh*	BBLV	Bokeloh bat lyssavirus	NC_025251
	*unclassified*	KBLV	Kotalahti bat lyssavirus	LR994545
II	*Lyssavirus lagos*	LBV a	Lagos bat virus/lineage a	EU293108
		LBV c	Lagos bat virus/lineage c	EF547449
	*Lyssavirus mokola*	MOKV	Mokola virus	EU293118
	*Lyssavirus shimoni*	SHIBV	Shimoni bat virus	NC_025365
III	*Lyssavirus caucasicus*	WCBV	West Caucasian bat virus	KY688150
	*Lyssavirus ikoma*	IKOV	Ikoma lyssavirus	NC_018629
	*Lyssavirus lleida*	LLBV	Lleida bat lyssavirus	MG983927
	*unclassified*	MBLV	Matlo bat lyssavirus	MW653808

**Table 3 viruses-15-00680-t003:** Results of mismatch identification between the sets of primers and probes and the retrieved lyssavirus sequences (n = 269).

TNMM *	rRT-PCR			RT-PCR		
	LN34		JW12/N165-146	Hemi-Nested		One-Step
	Primers	Probes		1st Round	2nd Round	
	*Phylogroup I (n = 251)*					
**0**	31%	99%	55%	70%	71%	70%
**1**	27%	1%	28%	29%	25%	29%
**2**	25%	0%	14%	1%	4%	1%
**3**	15%	0%	2%	0%	0%	0%
**4**	2%	0%	0%	0%	0%	0%
**5**	0%	0%	1%	0%	0%	0%
**6**	0%	0%	0%	0%	0%	0%
**7**	0%	0%	0%	0%	0%	0%
** *AC *** **	** *83%* **	** *100%* **	** *97%* **	** *100%* **	** *100%* **	** *100%* **
	*Phylogroup II (n = 12)*					
**0**	0%	50%	8%	8%	25%	25%
**1**	33%	50%	17%	83%	67%	58%
**2**	42%	0%	25%	0%	8%	0%
**3**	25%	0%	42%	8%	0%	17%
**4**	0%	0%	0%	0%	0%	0%
**5**	0%	0%	8%	0%	0%	0%
**6**	0%	0%	0%	0%	0%	0%
**7**	0%	0%	0%	0%	0%	0%
** *AC *** **	** *75%* **	** *100%* **	** *50%* **	** *91%* **	** *100%* **	** *83%* **
	*Phylogroup III (n = 6)*					
**0**	0%	83%	0%	33%	50%	0%
**1**	0%	17%	0%	50%	0%	83%
**2**	0%	0%	83%	0%	33%	17%
**3**	33%	0%	17%	17%	17%	0%
**4**	17%	0%	0%	0%	0%	0%
**5**	0%	0%	0%	0%	0%	0%
**6**	17%	0%	0%	0%	0%	0%
**7**	33%	0%	0%	0%	0%	0%
** *AC *** **	** *0%* **	** *100%* **	** *83%* **	** *83%* **	** *83%* **	** *100%* **

TNMM *: the total number of mismatches. AC **: acceptable complementarity. Percentage of sequences, displaying a TNMM of ≤ 2 with the primers or probes, as displayed in gray.

**Table 4 viruses-15-00680-t004:** List of primers (newly designed and existing) used in enhanced rRT-PCR.

Name	S/As	Sequence (5′ to 3′)	Length (nt)	Tm (°C)
FW1	S	ACGCTTAACAACMARATCAAAGAA	24	56.6–59.2
FW2	S	ACGCTTAACAACAAAATCADARAAG	25	55.7–58.6
FW3	S	ACGCTTAACGACAAAAHCAGARAAG	25	59.1–62.0
FW4	S	ACGCTTAACAGCTAAAAACYAGAAG	25	57.9–60.3
FW5	S	ACGCTTAACARCAAAATCTTATAAG	25	54.7–56.5
REV1	As	Same as original (LN34 REV) [26]	25	51.5–62.2
REV2	As	CTGGATATTTGTAYTCATAYTGATC	25	52.0–54.9
REV3	As	CAGGATATTTATATTCATACTGGTC	25	52.9
Probe1	S	Same as original (LN34 probe) [26]	17	48.7–51.6
Probe2	S	Same as original (LN34lago probe) [26]	17	45.8

S: sense, As: antisense. Data were generated by primer3 in Geneious Prime^®^ 2022.1.1 (Biomatters, Auckland, New Zealand).

**Table 5 viruses-15-00680-t005:** Summary of mismatches between the newly designed primer set and the 269 retrieved lyssavirus sequences.

TNMM *	Phylogroup I (n = 251)		Phylogroup II (n = 12)		Phylogroup III (n = 6)	
	Primers	Probes	Primers	Probes	Primers	Probes
**0**	58%	99%	0%	50%	83%	83%
**1**	37%	1%	50%	50%	0%	17%
**2**	5%	0%	25%	0%	17%	0%
**3**	0%	0%	25%	0%	0%	0%
**4**	0%	0%	0%	0%	0%	0%
**5**	0%	0%	0%	0%	0%	0%
**6**	0%	0%	0%	0%	0%	0%
**7**	0%	0%	0%	0%	0%	0%
** *AC* ** ** **** **	** *100%* **	** *100%* **	** *75%* **	** *100%* **	** *100%* **	** *100%* **

TNMM *: the total number of mismatches. AC **: acceptable complementarity. Percentage of sequences, displaying a maximum of two mismatches with the novel set of primers, as displayed in gray.

**Table 6 viruses-15-00680-t006:** Within-run and between-days repeatability tested on low and medium concentrations of EBLV-1 a (phylogroup I), LBV a (phylogroup II) and IKOV (phylogroup III).

Ph. *	RNA Transcript Dilution (GC/µL)	Within-Run Repeatability								Between-Days Repeatability			
		Day 1				Day 2							
		Mean	SD	CV	%	Mean	SD	CV	%	Mean	SD	CV	%
I	EBLV-1 a—low (1 × 10^2^)	32.8	0.70	0.02	97.87	32.3	0.59	0.02	98.17	32.6	0.63	0.02	98.05
	EBLV-1 a—medium (1 × 10^4^)	24.6	0.25	0.01	99.00	24.8	0.26	0.01	98.97	24.7	0.25	0.01	99.00
II	LBV a—low (1 × 10^2^)	32.5	0.86	0.03	97.37	32.1	0.75	0.02	97.67	32.3	0.75	0.02	97.66
	LBV a—medium (1 × 10^4^)	25.8	0.84	0.03	96.72	26.7	0.03	0.00	99.89	26.3	0.75	0.03	97.15
III	IKOV—low (1 × 10^2^)	32.7	0.70	0.02	97.85	33.6	0.53	0.02	98.44	33.2	0.73	0.02	97.81
	IKOV—medium (1 × 10^4^)	24.8	0.51	0.02	97.96	25.0	0.02	0.00	99.91	24.9	0.34	0.01	98.63

Ph. *—phylogroup, Mean—mean Ct values from three replicas, SD—standard deviation, CV—coefficient of variation, %—percentage of agreement, GC/µL—genome copies in µL.

## Data Availability

Not applicable.

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
