# Peer review of "Comparison of Pan-Lyssavirus RT-PCRs and Development of an Improved Protocol for Surveillance of Non-RABV Lyssaviruses"

_viruses, 2023, doi:10.3390/v15030680_

Round 1
Reviewer 1 Report
The manuscript submitted by Drzewnioková et al. presents a comparison between different molecular diagnostic tests based on RT-PCR and real-time RT-PCR (rRT-PCR) for the detection of all known lyssaviruses. The authors made an initial in silico evaluation of the expected performance of primers and probes, counting mismatches between all lyssaviruses described so far. Posteriorly, they tested the performance of these primers and probes with a set of RNA transcripts representing the four phylogroups of lyssaviruses and compared the Limit of Detection (LoD) for each diagnostic method. The authors further modified the primer set of one of the methods (LN34) to improve primer annealing by reducing the number of mismatches with respect to the total variability of lyssaviruses from the mentioned in silico analysis. This modified primer set showed increased inclusivity and proved to have enhanced sensitivity in detecting a wide number of strains compared to its previous formulation, as well as to other methods.
Overall, the manuscript is well-written and organized. It presents interesting results, since the authors provide an improved protocol for the diagnosis of lyssaviruses, with the broadest inclusiveness up to now. This was achieved by a meticulous assessment of both in silico analysis of mismatches against reference sequences, and testing the analytical sensitivity of each protocol. This novel method is a highly valuable resource that will benefit the detection of Lyssaviruses in diagnostic laboratories.
General comments:
Given the high levels of full complementarity observed for the conventional RT-PCR primers, is there a possibility to use them in a rRT-PCR protocol with SYBR Green? It could be interesting to discuss this possibility.
Specific comments:
Line 112: Explain the rationale for the selected threshold of two mismatches.
Line 119: I understand that the authors did not produce the full combinatory representing all possible combinations of ambiguous sites, and instead they synthesized observed combinations derived from the MSA. Whether this interpretation is correct or not, please clarify this point.
Line 153: The JW12/N165-146 was not used with TaqMan probes (Table 1, section 2.5). This should be explicit. The authors should further clarify if they used SYBR Green for establishing the LoD.
Line 168: Please explain the rationale for the use of this threshold (Ct values up to 35.9).
Line 169: Please clarify the method used to analyze melting temperature and amplification curves.
Line 170: This decision (the highest dilution at which the amplification products could be confirmed by Sanger sequencing) may underestimate the LoD of this method. It is frequent to obtain conspicuous gel bands, and not necessarily a good-quality sequence. It would be interesting if the authors could also compare the results obtained using a LoD based on PCR band observation. If the authors are not able to reproduce these results, it would be desirable that they problematize this decision.
Line 191: Please revise Figure numbering and in-text figure references throughout the manuscript (and in figure legends).
Line 201: Based on which results? The conventional RT-PCR has fewer mismatches than LN34.
Line 208: Please revise figure numbering.
Line 209: This is true for the original LN34 protocol. But the conventional RT-PCR still has a lower number of mismatches. What would be the advantage of using a TaqMan method, vs adapting the conventional RT-PCR to a rRT-PCR using SYBR Green?
Lines 236 and 295: Again, this is influenced by the selected threshold (Sanger sequence obtention). It should be mentioned.
Other minor comments and grammar suggestions are included in the attached file.

Reviewer 2 Report
The manuscript entitled Comparison of pan-lyssavirus RT-PCRs and development of an improved protocol for non-RABV lyssaviruses surveillance written by Drzevnikova et al. presents a very valuable data concerning molecular protocols of lyssaviruses detection. The study is well designed and concerns on-going evaluation of molecular protocols due to genetic diversity of lyssaviruses, potential generation of a novel lyssavirus species and its potential spillover in the fields.
The study contains in silico evaluation of pan conventional and real-time RT-PCRs broadely used over the world. While real-time RT-PCR protocols compared in the study are recommended both by WHO and WOAH organisations for molecular detection of lyssaviruses, conventional PCR is presented by DeBenedictis et al. protocol that is not as wide used as that recommended by WHO and WOAH. To increase the value of the manuscript I highly recommend to extend the in silico evaluation as well as the validation and comparison of the newly developed primers (n) LN 34 to Heaton et al. protocol, recommended by WHO and WOAH for pan-lyssavirus detection using heminested RT-PCR protocol, that in many countries is still the only molecular protocol. If the authors performed such kind of studies/ comparisons it should be added it the text.
In Materials and Methods section it should be highlited that Weakeley primers ( ref. 26) are implemented as OneStep RT-PCR protocol with SYBR Green detection.
Please, veryfy references 18 and 23? In my opinion, the reference is duplicated, unless it refers to other pages in the manual. In my opinion, the reference is duplicated, unless it refers to other pages in the manual. In general, I recommend to check the references under the editing manner.
Reviewer 3 Report
Lyssavirus contains more than 17 species, which have a broad host range. Therefore, it is urgent to have a reliable and rapid diagnostic method for monitoring lyssavirus in animals and humans. The manuscript entitled “Comparison of pan-lyssavirus RT-PCRs and development of an improved protocol for non-RABV lyssaviruses surveillance” by Drzewnioková1 et al presents an improved real-time RT-PCR method for detection of pan-lyssavirus and compared it with three widely used molecular diagnostic methods. This study is well planned and performed, but these are some details need to revise as follows:
1. As internationally recommended and widely used methods, is there any deficiency or limitation in LN34 or JW12/N165-146 assay? Relevant information needs to be provided in the “Introduction”.
2. There is an extra half bracket on line 121.
3. Line 133, “Plasmids containing a lyssavirus genome sequence encompassing assay target regions ……” . How to confirm the size and integrity of RNA transcribed in vitro? More accurate methods such as ddPCR should be used to calculate the amount of standard templates.
4. Line 170-171, “For conventional RT-PCR, …… the amplification products could be confirmed by Sanger sequencing.” It seems inconvenient to judge the amplification result. Why and how to use Sanger sequencing?
5. Some ways should be used to prove the specificity and repeatability of (n) LN34 assay.
6. “……, which encompassed 20 and 4 forward and reverse primers, respectively.” In line 326, “……and the 2002 sequence (EF614258)……” in line 338, “……but the entire target region of the LN34 assay,…… ” in line 340, these sentences are confusing.
7. Line 329, “……(n) LN34 assay resulted in 100% complementarity with……”. Obviously, it is not 100% complementary.
8. Optimize and simplify the “Conclusions”.
Round 2
Reviewer 3 Report
The author's reply and modification can well remove my previous doubts.
